# The Role of DND1 in Cancers

**DOI:** 10.3390/cancers13153679

**Published:** 2021-07-22

**Authors:** Yun Zhang, Jyotsna D. Godavarthi, Abie Williams-Villalobo, Shahrazad Polk, Angabin Matin

**Affiliations:** Department of Pharmaceutical Sciences, College of Pharmacy and Health Sciences, Texas Southern University, Houston, TX 77004, USA; j.godavarthi1424@student.tsu.edu (J.D.G.); a.williams8561@student.tsu.edu (A.W.-V.); s.polk7032@student.tsu.edu (S.P.)

**Keywords:** DND1, germ cell, teratomas, somatic cancers, translation regulator

## Abstract

**Simple Summary:**

The Dead-End (DND1) protein can interact with different messenger RNAs (mRNAs) in the cell. It uses multiple mechanisms to regulate expression of proteins from their cognate mRNAs. High levels of DND1 are found in the progenitor cells that develop into the egg and sperm. Here we review how and why defects in DND1 cause tumors in the testes and ovaries of vertebrates. Unexpectedly, some recent reports indicate that DND1 may also participate in human cancer development in cells other than those of the testes and ovaries. The goal of this review is to summarize the literature on the role of DND1 in cancers to obtain perspective regarding future scientific endeavors on DND1 function.

**Abstract:**

The *Ter* mutation in Dead-End 1 (*Dnd1*), *Dnd1^Ter^*, which leads to a premature stop codon, has been determined to be the cause for primordial germ cell deficiency, accompanied with a high incidence of congenital testicular germ cell tumors (TGCTs) or teratomas in the 129/Sv-*Ter* mice. As an RNA-binding protein, DND1 can bind the 3′-untranslated region (3′-UTR) of mRNAs and function in translational regulation. DND1 can block microRNA (miRNA) access to the 3′-UTR of target mRNAs, thus inhibiting miRNA-mediated mRNA degradation and up-regulating translation or can also function to degrade or repress mRNAs. Other mechanisms of DND1 activity include promoting translation initiation and modifying target protein activity. Although *Dnd1^Ter^* mutation causes spontaneous TGCT only in male 129 mice, it can also cause ovarian teratomas in mice when combined with other genetic defects or cause germ cell teratomas in both genders in the WKY/Ztm rat strain. Furthermore, studies on human cell lines, patient cancer tissues, and the use of human cancer genome analysis indicate that DND1 may possess either tumor-suppressive or -promoting functions in a variety of somatic cancers. Here we review the involvement of DND1 in cancers, including what appears to be its emerging role in somatic cancers.

## 1. Introduction

The story of Dead-End 1 (DND1) in cancer of mouse germ cells can be traced back almost half a century. A new spontaneously arising mutation led to generation of the new inbred subline of mice (129/Sv-*Ter*) that exhibited primordial germ cell (PGC) deficiency, accompanied with a high incidence of congenital testicular germ cell tumors (TGCTs) or teratomas [1,2,3]. These recessive phenotypes mapped to mouse chromosome 18, or the *Ter* locus [4,5]. In 2005, twenty years after the 129/Sv-*Ter* mouse line was isolated, positional cloning of *Ter* identified the genetic defect to a mutation in *Dead-End* 1, a mouse ortholog of the zebrafish *dead-end* gene required for PGC migration and survival [6]. Mouse *Dnd1* contains four exons. *Ter* is due to a single nucleotide substitution occurring spontaneously in exon 3 of *Dnd1*, which transforms the arginine residue at amino acid (aa) 190 into a premature stop codon, thus causing either truncation or loss of DND1 expression (Figure 1). This mutation results in the phenotypes of PGC loss, male and female sterility, and the high incidence of TGCT in males of the 129/Sv mice.

DND1 is an RNA-binding protein that contains two RNA recognition motifs (RRMs) in tandem, spanning approximately aa residues 58–136 and 138–218 (https://www.uniprot.org/uniprot/Q6VY05, accessed on 18 July 2021), respectively, followed by a double stranded RNA-binding motif at the carboxyl (C)-terminus (Figure 1). The amino (N)-terminal RRM1 is canonical and essential for binding specific mRNAs [7,8]. The RRM2 is less conserved and contains the HRAAAMA motif (Figure 1, described below). Over the years, DND1 has been found to possess diverse molecular functions. The most studied is its role in translation regulation [7,8,9,10]. In addition to TGCTs, DND1 has also been shown to potentially impact ovarian teratomas (OTs) and somatic tumors [11,12,13,14,15,16,17,18,19]. Because DND1 has been described as a protein primarily expressed in germ cells, the multiple research reports and database evidence of DND1 involvement in somatic cancers are unexpected. Here we review studies that have investigated mechanisms of DND1 function in vertebrates and especially its role in somatic cancers, including human cancers. A PubMed search using “DND1” as the keyword was conducted, which returned 121 results as of June 2021. These 121 papers were manually screened to identify those that are related to the topic of this present review.

## 2. Molecular Mechanisms of DND1 Function

### 2.1. mRNA Stabilization

Several studies have demonstrated that DND1 can bind specific mRNAs and block microRNA (miRNA) access from the 3′-untranslated region (3′-UTR) of target mRNAs and inhibit miRNA-mediated mRNA degradation, thus up-regulating translation (Figure 2a). The target mRNAs of DND1 that have been studied are often those involved in regulation of the cell cycle or apoptosis. For example, DND1 binds to the 3′-UTR of *p27* mRNA and blocks miR-221 so as to up-regulate p27 protein expression [7]. Increase in the level of this cyclin-dependent kinase inhibitor is well known to decrease cell proliferation and induce apoptosis [20]. Uridine-rich regions (URRs) on the 3′-UTR of the *p27* transcript mediate the DND1-3′-UTR interaction, as mutating the URRs impaired DND1-mediated up-regulation of *p27* translation [7]. In another example, zebrafish Dnd1 protein binds the URR in the 3′-UTR of the mRNA of the DNA replication inhibitor, *geminin*, and increases its translation [21]. In addition to directly regulating cell cycle via stabilizing *p27* or *geminin*, DND1 can also impose indirect regulation of the cell cycle. For example, through blocking miR-26a from enhancer zeste homolog 2 (*EZH2*) mRNA that encodes a methyltransferase responsible for histone H3 lysine 27 (H3K27) trimethylation (me3), DND1 up-regulates the translation of *Ezh2* [22]. In turn, EZH2 protein represses cyclin *CCND1* expression to inhibit entry into the cell cycle.

During *Xenopus* embryogenesis, Dnd1 has been reported to bind to the 3′-UTR of *trim36*, which is essential for vegetal cortical microtubule assembly, promoting localized high concentration of Trim36 protein in the vegetal cortex in the *Xenopus* egg. Thus, Dnd1 also appears to have an essential role in vegetal cortical microtubule assembly during axis specification during embryogenesis [23].

DND1 function of blocking miRNA access is also subject to negative regulation or DND1 itself is subject to miRNA regulation. For example, *Xenopus dnd1* mRNA can be cleared by *miR-18* in the soma and specifically protected from degradation in germ line of *Xenopus* embryos via Elr-type proteins [24]. Another example in squamous cell carcinoma cell lines demonstrated that miR-24 down-regulates DND1 [13]. The DNA cytosine deaminase, apolipoprotein B mRNA-editing enzyme, catalytic polypeptide-like 3 (APOBEC3), was found to interact with DND1 [25] and counteract DND1 function by restoring the suppressive activity of miRNA on *p27*, *LATS2* and *CX43* mRNAs [26].

A recent study dissected the individual functions of the two RRMs of Dnd1 in regulating *nanos1* translation in *Xenopus* germline [8]. The N-terminal RRM1 is critical for interaction with mRNA targets [7,8,10]. The RRM2 possesses ATP-dependent helicase activity [8,27] required to resolve the secondary structure of bound mRNAs to promote translation [8]. The above-mentioned HRAAAMA motif is presumably part of the putative ATPase domain. Furthermore, CRISPR-Cas9-mediated base-editing generated mutations of four aas in RRM1 of mouse DND1 (E59, V60, P76 and G82), which functions in mRNA binding, depleted PGCs in mouse embryos, indicating the essential role of these RRM1 aas for DND1 function during PGC development [28].

### 2.2. mRNA Repression or Degradation

Other studies show that DND1 can also function as a translation suppressor [9,10,29]. It binds a UU(A/U) trinucleotide motif predominantly in the 3′-UTR regions of mRNA and in addition recruits the CCR4-NOT deadenylase complex, to degrade target mRNAs (Figure 2b). The mRNA targets of DND1 destabilized by CCR4-CNOT include genes associated with apoptosis, inflammation, and pathways that regulate stem cells, such as of the TGF, WNT and PI3K-AKT signaling pathways [10]. In germ cells, DND1 has also been shown to partner with NANOS2 [9,30,31], an evolutionarily conserved RNA-binding protein specifically involved in germ cell development, and then interact with CCR4-NOT. DND1 is required for loading unique RNAs into the Nanos2-CCR4-NOT complex [9], likely accounting for the synergistic increase of incidence of TGCT in double mutants for DND1 and NANOS2 [31]. On the other hand, Dnd1 has also been shown to bind mRNA and repress their translation without affecting their stability during zebrafish embryogenesis [29]. Further studies to elucidate the exact cellular and physiological context of these mechanisms are needed.

DND1 has a long list of target mRNAs [10,23,32,33,34], which is expected to grow longer. In germ cells, DND1 down-regulates many genes associated with pluripotency and active cell cycle (e.g., *Rheb*, *Rhob*, *Ccne1*, and *Trp53* etc.) whereas it up-regulates chromatin regulator genes (e.g., *Kat7*, *Rbbp5*, *Kdm5a*, and *Setdb1* etc.) [34], suggesting that post-transcriptional up- or down-regulation may exist concurrently in cells and could be mRNA target-specific.

### 2.3. Other Molecular Functions of DND1

In addition to overcoming the inhibitory action of miRNA, DND1 can also activate germline-specific translation through promoting initiation. In *Xenopus* oocytes, Dnd1 binds to the mRNA of germline gene *nanos1* and interacts with eukaryotic initiation factor 3f (eIF3f) which is a repressive component in the 43S preinitiation complex [35]. This Dnd1-eIF3f interaction counteracts the repressive activity of eIF3f and leads to *nanos1* translation.

DND1 was also found to possess a unique function of modifying the activity of target protein. In mouse embryos, DND1 interacts with the activator protein 1 (AP1) subunit c-Jun [36]. Co-transfection with *Dnd1* and *c-Jun* plasmids in GC-1 mouse spermatogonia cells significantly increased the transcriptional activity of AP1.

DND1 has been detected in both the cytoplasm and nucleus [37,38,39], in line with the function of DND1 as a post-transcriptional [7,9,10,29] or transcriptional regulator [36]. What determines the multifaceted regulatory roles of DND1 is still unknown. Whether the different reported functions of DND1 are species-specific also remains to be clarified. A recent study revealed an interesting 3D domain swapped dimerization of the DND1-RRM2 domain [40], which increases surface area for multimeric interactions and may allow DND1 to exert different functions in different context. Further investigations are warranted to elucidate the mechanisms and causes underlying the multifaceted roles of DND1.

## 3. *Dnd1^Ter^* Mutation in Testicular and Ovarian Teratomas

In vertebrates, PGCs segregate from pluripotent epiblast cells and migrate into the developing gonads, where they specify into either male or female germ cells. In mice, *Dnd1* expression is specifically induced in PGC precursors at embryonic day (E) 6.5–6.75 and maintained until the pre-meiotic spermatogonia stage in adult testes [10]. Immunoblotting using mouse-specific anti-DND1 antibodies has detected DND1 in the spermatids of adult mouse testes but not in Sertoli cells [39]. Additionally, examination of The Human Protein Atlas indicates *DND1* mRNA is enriched in spermatogonia and early spermatids (https://www.proteinatlas.org/ENSG00000256453-DND1/celltype, accessed on 18 July 2021) from human testis [41,42,43].

The *Dnd1^Ter^* mutation causes PGC loss in all mouse genetic backgrounds, leading to infertility in both genders. However, only male *Dnd1^Ter^* mice on the 129 strain background develop the cancerous TGCT. In these mice, some of the remaining PGCs acquire the intermediate pluripotent status [44] to transform into highly proliferative, undifferentiated pluripotent embryonal carcinoma (EC) cells. At the time of birth, most EC cells differentiate into random mix of various cells and tissues that compose the teratomas of TGCTs. Although the incidence of teratomas in wild-type (WT) strain 129/Sv males is 1.4%, the *Dnd1^Ter^* mutation increases teratoma incidence to 17% in heterozygous mutants and 94% in homozygotes [2]. Thus, *Dnd1^Ter^* is a potent modifier of spontaneous TGCT susceptibility on the 129 strain background.

More strikingly, in the WKY/Ztm rat strain, homozygous mutation in *Dnd1* led to formation of congenital testicular and ovarian teratomas, as well as infertility with complete penetrance in both genders [11]. The rat *Dnd1^Ter^* was determined as a point mutation in exon 4 of rat *Dnd1*, which introduces a premature stop codon at aa 289, resulting in a longer truncated protein compared to the predicted mouse *Dnd1^Ter^* mutant (Figure 1). The HRAAAMA motif is lost in the mouse DND1^Ter^ mutant due to the premature stop codon at aa 190, but remains intact in rat Ter [14] (Figure 1), which may explain for the gender difference in Dnd1*^Ter^*-related teratocarcinogensis between mouse and rat. A series of studies from Tokumoto’s lab also point to the role of DND1 in female germ cells. They determined that the *ett1* locus, on mouse chromosome 18 but separate from the *Ter* locus, is responsible for experimentally testicular teratoma (ETT) formation induced in 129/Sv mice when E12.5 fetal testes are transplanted into adult testes [45]. They subsequently identified a single nucleotide polymorphism that introduces glycine to serine mutation at aa codon 25 (G25S) in melanocortin 4 receptor (*Mc4r*) gene within the *ett1* locus [46]. In a very recent report, they showed that the MCR4 G25S homozygotes also caused OTs) in 48.7% of LTXBJ mice. The incidence of OT was further increased to 66.7% when *Ter* locus of the 129 strain was introduced in heterozygous to *Mc4r^G25S/G25S^* mice [12].

DND1 was found to interact with several pluripotency associated mRNAs, *OCT4*, *SOX2*, *LIN28* and *BCL2L1*, in the germinal vesicle-stage of pig oocytes [47]. In *Xenopus* oocytes, Dnd1 was also subject to ubiquitin-independent proteasomal degradation and translational repression, regulations that prevent premature accumulation of Dnd1 in oocytes [48]. These reports highlight the role of DND1 and its interactions with specific genetic susceptibility factors that result in germ cell tumors (TGCTs or OTs) in male and female animal model systems.

In contrast to its function in mouse TGCT, germline *DND1* mutations were shown unlikely to contribute significantly to human TGCT susceptibility [49,50]. In addition, DND1 was not detected among the panel of putative prognostic and diagnostic markers of human TGCT tissues identified by proteomic techniques [51], indicating DND1 may not play a significant role in human TGCT. The functional discrepancy between mouse and human DND1 in TGCT is probably not surprising, considering that even in mouse, the increase of TGCT incidence by *Dnd1^Ter^* is highly dependent on the strain genetic background [32] and genetic susceptibility factors other than *Dnd1^Ter^* can also increase TGCT occurrence in mouse [45,52,53]. On the other hand, mechanistic studies indicate a link between DND1 and miRNAs that characterize human TGCTs. For example, DND1 was found to interact with *LATS2* mRNA to alleviate miR-372-mediated *LATS2* suppression [7,32,33]. Furthermore, proteins such as APOBEC3 that interact with DND1 can modulate DND1 function of blocking miR-372 [26]. Significantly, miR-372 belongs to the miR-371-373 cluster, which is one of the most relevant biomarkers for TGCTs in patients [54,55]. Thus, further investigation is warranted to elucidate any putative link between DND1 function and miRNA signatures of human TGCT.

## 4. The Emerging Role of DND1 in Somatic Cancers

Although in the zebrafish, Dnd1 is exclusively expressed in germ cells, in mammals it appears that lower levels of DND1 are expressed in a variety of cells and tissue types, mainly of epithelial or mesenchymal origin. DND1 expression has been reported in normal human neutrophils, hematopoietic CD34+ progenitor cells [16], rat neural stem cells, neurons and astrocytes [56] and in breast epithelial cells [17]. Additionally, emerging evidence indicates a role of DND1 not limited to only in germ cell cancers but also in some types of somatic cancers [13,14,15,16,17,18,19]. Table 1 summarizes the studies of DND1 in different cancer types. Importantly, these studies on DND1 not only used established cancer cell lines but used patient derived cancer tissue samples. We discuss below the studies on DND1 in breast cancer, gastrointestinal (GI) cancers, tongue squamous cell carcinoma (TSCC) and leukemia.

In a study on human breast cancer by Cheng et al., the group’s analysis of The Cancer Genome Atlas (TCGA) microarray mRNA expression profiles for breast cancer found that patients with higher *DND1* levels had longer overall survival [17]. Furthermore, comparison of 21 pairs of breast tumors and their neighboring mammary normal epithelial tissues found two-fold lower *DND1* mRNA levels in breast cancer tissues compared to adjacent normal breast tissues. The lowest *DND1* levels were detected in a panel of human breast cancer cell lines with highest metastatic capability. In addition, *DND1* expression was found to be positively correlated with the pro-apoptotic effector *BIM* expression in human breast cancer. Experimentally, knockdown of *DND1* in MCF-7 cells decreased BIM expression and inhibited apoptosis. DND1 was found to increase BIM expression and stability by blocking miR-221 from *BIM*-3′ UTR.

Several studies also indicate that DND1 functions in GI cancers. In hepatocellular carcinoma (HCC) cells, DND1 overexpression was found to inhibit spheroid formation, suppress HCC cancer cell stemness, inhibit epithelial-mesenchymal transition and increase the sensitivity of HCC cells to sorafenib [18]. The tumor-suppressive function of DND1 in HCC may at least partially result from Hippo signaling activation. It was shown that through binding to *LATS2* 3′-UTR, DND1 overexpression led to elevated LATS2 level and YAP phosphorylation and retention in the cytoplasm, diminishing the transcriptional activity of the YAP oncogene.

In another study, mutant *Dnd1^Ter^* was found to significantly increase polyp number and mass in the *Apc^+/Min^* mouse model of intestinal polyposis [14], indicating the tumorigenic properties of *Dnd1^Ter^* in the intestine. These above studies indicate that normal, WT DND1 has tumor-suppressive type activity with increased DND1 expression being anti-proliferative, pro-apoptotic, or causing reduction of cancer stemness. However, one study contradicts this in that DND1 seems to play a tumor-promoting role in SW48 colorectal cancer cell line [19], as silencing DND1 suppressed cell proliferation and overexpression of DND1 reversed the tumor-suppressive effects of miR-24.

In TSCC cells, DND1 itself was also identified as a target of miR-24 [13]. Down-regulation of DND1 mediated by miR-24 led to reduced expression of cyclin-dependent kinase inhibitor 1B (CDKN1B), whose translation is up-regulated by DND1. This was accompanied by enhanced cell proliferation.

Lastly, DND1 seems to also play a role in non-solid tumors. The expression of DND1 and another RNA-binding protein RBM binding motif protein 38 (RBM38) is repressed in primary acute myeloid leukemia patients [16]. Inhibition of *DND1* and *RBM38* mRNA significantly attenuated differentiation of NB4 acute promyelocytic leukemia cells and resulted in decreased *p21(CIP1*) mRNA expression.

In summary, different studies on DND1 function found that DND1 is an anti-proliferative, pro-apoptotic tumor suppressor in a variety of cancers, but may also exert oncogenic function in others. The presence of different repertoires of mRNA/protein in different cells or tissues may account for the opposite roles of DND1 detected in the cancer studies, as DND1 stabilizes some mRNAs but suppresses or degrades others [7,9,10,29] (Figure 2), and can also modify the activity of target protein [36]. Moreover, how DND1 functions under different physiological conditions in the cell is unknown. It is also noteworthy that the above-described studies in different cancer types each represent a single study and further investigations are critically needed to verify these findings.

To complement the above-mentioned experimental studies on the role of DND1 in somatic cancers, we queried the frequency of *DND1* gene alteration in human cancers from TCGA database. We examined data on liver, breast, skin, colon neoplasms and AML in cBioPortal (cbioportal.org, accessed on 30 June 2021) [57,58]. *DND1* gene shows minimal alteration in these cancer types that we analyzed (data not shown), except that it is altered, mostly through deletion, in 9.38% of melanomas. In addition, we found high *DND1* alteration frequency (>5%) in other three cancer types: lung (up to 18.75%), prostate adenocarcinoma (up to 6.49%), and kidney renal clear cell carcinoma (up to 14.12%). Interestingly, while the majority of *DND1* alteration is deletion or mutation in lung and prostate cancers, *DND1* amplification prevails in kidney cancers. It is imperative to further explore the role of DND1 in these cancer types as to whether the changes in *DND1* represent passenger events or whether DND1 directly contributes to cancer susceptibility and progression.

## 5. Genetically Engineered Mouse Alleles of *DND1*

In contrast to the cancer cell line studies and cancer genome data analysis, in vivo model systems allow control of genetic background and testing of direct causal relationship, and thus will serve as important tools for elucidating the possible roles of DND1 in somatic cancers. In addition to the above-mentioned mice carrying the mutations of four aa residues in RRM1 of DND1 (E59, V60, P76 and G82) [28], hereto, several genetically engineered mouse alleles of *Dnd1* have been generated by different laboratories to study germ cell biology and germ cell cancers.

Two *Dnd1* knockout mouse alleles generated by different research groups have reported to manifest different phenotypes and thus whether *Ter* simply causes DND1 loss or produces a truncated DND1 protein remains controversial. The conventional *Dnd1*-knockout mouse line, *Dnd1∆* [31], was established by administering tamoxifen globally to *Dnd1^+/flox^*; *Rosa26^+/CreERT2^* female mice [9]. These *Dnd1∆* mice showed phenotypes similar to those of *Ter* mutant mice in spermatogenesis, oogenesis, and teratoma incidence, with a slight difference in spermiogenesis. The DND1 protein level in *Dnd1^+/Ter^* mice was half of that in WT embryos and the expression of the shorter, truncated DND1 protein was not detected. These data support that *Ter* mutation causes DND1 loss. However, another study indicated that *Dnd1^Ter^* is not functionally equivalent to DND1 loss in mice [14]. This *Dnd1* knockout allele, *Dnd1^KO^*, caused embryonic lethality in *Dnd1^KO/KO^* homozygotes and *Dnd1^+/KO^* heterozygotes were born with reduced numbers. *Dnd1^Ter^* only partially rescued these phenotypes. In addition, while *Dnd1^+/KO^* heterozygous male mice did not exhibit increased occurrence of TGCT, a single copy of *Dnd1^Ter^* was able to increase TGCT risk regardless of whether the alternative allele was loss-of-function (*Dnd1^KO^*) or WT. The phenotypic discrepancy of the *Dnd1∆* and *Dnd1^KO^* mice may at least be partially due to the slight genetic difference between 129 substrains. In addition, the fact that different portions of the *Dnd1* gene were deleted in these two alleles may also account for the discrepancy. In *Dnd1∆* allele, exons 2-3 were removed, whereas exons 1–2 and most of exon 3 of *Dnd1* were deleted in the *Dnd1^KO^* allele. One cannot exclude the possibility that a truncated DND1 is being produced from the *Dnd1^KO^* allele that retains partial DND1 function and could thus reduce TGCT occurrence.

Mouse alleles that overexpress DND1 have also been generated. The *LSL-FM-Dnd1^flox^* ‘knock-in’ allele allows *Cre recombinase*-mediated expression of genetically engineered, FLAG- and myc-tagged DND1 in a cell-type-specific manner [59]. The role of this allele in cancer remains to be tested. The *Dnd1-3XFLAG* transgenic allele contains 3×FLAG-tag encoding sequence at the C-terminus of *Dnd1*, resulting in the expression of DND1-3×FLAG fusion protein under the direct control of the *Dnd1* enhancer [9]. As good antibodies for co-immunoprecipitation experiments for DND1 are still unavailable, these alleles allow pull down of DND1 binding partners with FLAG or myc antibodies. Furthermore, the newly reported transgenic mouse line carrying a *Dnd1GFP* fusion allele enables in vivo imaging of the DND1 protein and easy sorting of DND1 expressing cells [60].

Introduction of these DND1 deleted or overexpressing alleles into mouse models for different cancer types will be highly informative to determine whether and how DND1 deletion or overexpression affects somatic cancers. These mouse alleles will undoubtedly serve as valuable tools for addressing the role of DND1 in vivo and advance our knowledge of DND1 function.

## 6. Conclusions and Future Perspective

Various studies have revealed and are still unravelling the molecular mechanisms of DND1 function and its role in human cancer. Not surprisingly, DND1, as with many other RNA-binding proteins, has multiple functions as demonstrated in the early embryo, germ cells as well as in some mammalian somatic cells. DND1 has been found to regulate gene translation through various mechanisms, including blocking miRNA access to target RNA to stabilize the RNA, recruiting the CCR4-NOT complex to degrade the target RNA, and activating preinitiation complex to promote translation initiation.

Several questions remain unresolved in the rodent system, such as whether DND1*^Ter^* is expressed and how and whether it acquires additional gain-of-function properties. In primordial germ cells of the mouse, WT DND1 interacts with germ cell-specific factors, for example NANOS2 [9,30,31], to facilitate germ cell progression through rapidly changing developmental states. As such, DND1 maintains cell viability while facilitating changes in the pluripotency status and cell cycle states of germ cells. The exact mechanisms through which DND1 functions to maintain viability of early germ cells is unclear. Defects in DND1, such as *Dnd1^Ter^* mutation, lead to germ cell depletion and high TGCT incidences in genetically susceptible rodent strains where DND1*^Ter^* appears to function as a cancer modifier.

In addition to germ cell tumors, DND1 has been shown to be involved in some somatic cancers. Thus far most experimental studies in somatic cancer cells indicate that WT DND1 functions as a tumor suppressor with anti-proliferative and pro-apoptotic effects. Cellular proteins that interact with DND1 to facilitate its function in somatic cells remain to be discovered.

Human cancer database analysis revealed *DND1* gene alterations in several different cancer types. Whether DND1 is expressed ectopically in these cancers or induced due to some stress response in cancer cells remains to be examined. Interestingly, in neuronal cells, DND1 expression is turned on due to alcohol induced stress, most likely to facilitate apoptosis [56,61]. Thus, we speculate that certain somatic cells may also retain the capacity to induce DND1 in response to, for example, oncogenic stress. These ideas could be tested experimentally using cell lines and animal models.

Compared with other extensively studied cancer-related genes, such as *TP53*, *PTEN*, and *KRAS*, etc., the history of *DND1* is still relatively short and its role in different cancers is largely unknown. Key unanswered questions about DND1 include, but are not limited to, what causes the 129-mouse strain-specific high TGCT incidences in *Dnd1^Ter^* mice, what mechanisms steer the translation activating or suppressive role of DND1, and how DND1 functions in cancer cells. Understanding the myriad of DND1 functions, aided by animal models, remains a critical avenue for future studies, as it is through these mechanisms that we may come to understand cancer susceptibility factors to rationally design anti-cancer agents.

## Figures and Tables

**Figure 1 cancers-13-03679-f001:**
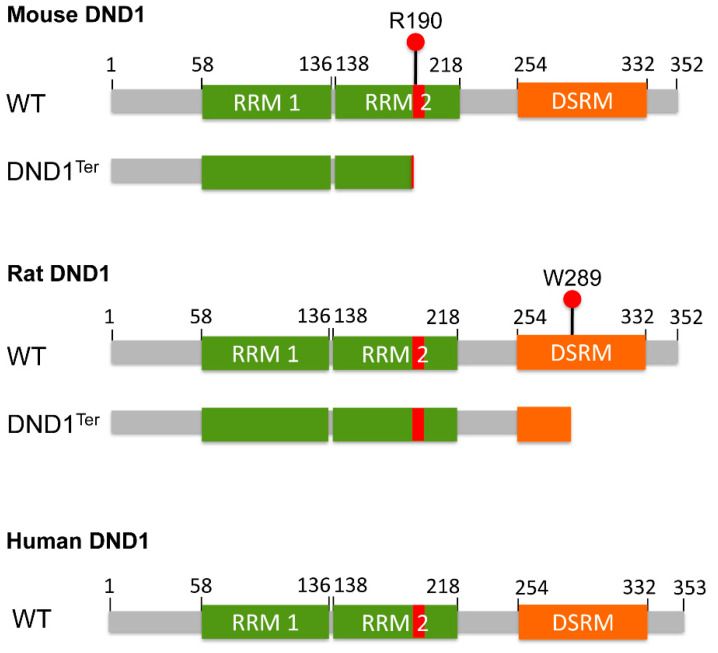
Schematic representation of the structure of mouse, rat, and human DND1 proteins. The RNA recognition motif 1 (RRM1), RNA recognition motif 2 (RRM2) and double stranded RNA-binding motif (DSRM) are shown. *Ter* mutation transforms arginine at amino acid (aa) 190 in mouse and tryptophan at aa 289 in rat DND1 into a premature stop codon. The red bar in RRM2 represents the HRAAAMA motif spanning from aa 189 to 195 in all three species. The predicated mouse and rat DND1^Ter^ protein are also shown underneath the wild-type (WT) protein, respectively. Please note that whether DND1^Ter^ causes truncation or loss of DND1 expression is still controversial. The NCBI Reference Sequence numbers for mouse, rat and human DND1 proteins are NP_775559.2, NP_001102849.1 and NP_919225.1, respectively, accessed on 20 July 2021.

**Figure 2 cancers-13-03679-f002:**
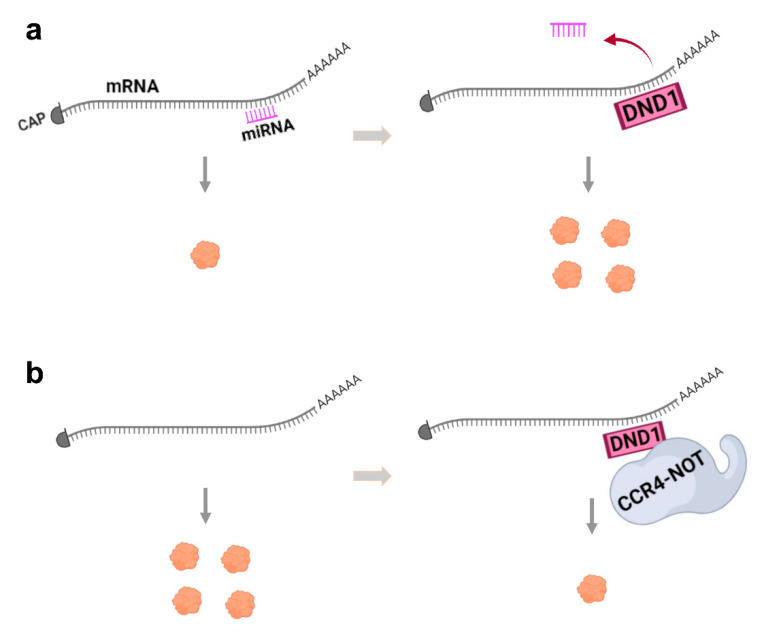
The two major mechanisms of DND1 function. (**a**) Interaction of DND1 with mRNA blocks miRNA access to mRNA to up-regulate translation and protein expression. (**b**) DND1 can also bind to mRNA to target it for destruction by the CCR4-NOT complex, thus decreasing translation and protein expression. These mechanisms may be cell-type- or mRNA-specific.

**Table 1 cancers-13-03679-t001:** Studies on the role of DND1 in somatic cancers.

Tumor Type	Endogenous DND1 Status in Human Cell or Tissue Samples	Phenotypes Caused by Experimental Alteration of DND1	Mechanism of DND1 Function	Reference
Breast cancer	Lower expression of *DND1* levels in breast cancer tissue compared to normal.	Knockdown of DND1 in MCF-7 cells decreased BIM expression and inhibited apoptosis.	DND1 increases expression of BIM by blocking miR-221 from *BIM*-3′UTR.	[17]
Hepatocellular carcinoma (HCC)	DND1 mRNA and protein levels significantly decreased in HCC sphere cells.	DND1 overexpression inhibited spheroid formation; suppressed HCC cancer cell stemness; inhibited epithelial-mesenchymal transition; increased sensitivity of HCC cells to sorafenib.	DND1 binds to LATS2 3′-UTR, elevating LATS2 level and YAP phosphorylation and retention in the cytoplasm, therefore diminishing transcriptional activity of YAP.	[18]
Intestinal polyposis	N/A	*Apc^+/Min^Dnd1^+/Ter^* mice had higher polyp numbers compared to *Apc^+/Min^Dnd1^+/+^ mice.*	N/A	[14]
Colorectal cancer (CRC)	DND1 expression significantly up-regulated in CRC cell lines.	Silencing DND1 reduced SW48 cell line viability and overexpression of DND1 promoted cell proliferation.	DND1 is the potential target of miR-24 in SW48 cells and involved in miR-24 mediated inhibitory effects on cell proliferation.	[19]
Tongue squamous cell carcinoma (TSCC)	Reduced expression of DND1 in TSCC cells and tissues.	DND1 knockdown in TSCC cell lines enhanced cell proliferation and reduced apoptosis. Enhanced DND1 expression reduced cell proliferation and increased apoptosis.	DND1 is a direct target of miR-24. miR-24 suppressed DND1, leading to reduced CDKN1B.	[13]
Acute myeloid leukemia (AML)	Lower *DND1* mRNA levels in AML blasts and CD34+ progenitor cells.	Inhibition of both *RBM38* and *DND1* mRNAs significantly attenuated NB4 differentiation and resulted in decreased *p21(CIP1*) mRNA.	Activity of RBM38 and DND1 during neutrophil differentiation antagonize the activity of oncomiRs to protect mRNAs, for example *p21CIP1* that are important for myeloid differentiation.	[16]
Skin cancer	Reduced expression of *DND1* mRNA in transformed HaCaT cells and loss of *DND1* mRNA and protein in tumors from transformed HaCaT cells.	Expression of DND1 in transformed HaCaT cells interfered with miR-21-mediated repression of MSH2; Knockdown of DND1 reduced *MSH2* RNA, an effect further enhanced by miR-21.	DND1, which decreases sensitivity of *MSH2* to miR-21, is down-regulated during tumorigenesis therefore increasing the effectiveness of miR-21 in tumors.	[15]

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
