# Peer review of "The Role of DND1 in Cancers"

_cancers, 2021, doi:10.3390/cancers13153679_

Round 1

Reviewer 1 Report

In this manuscript, the authors comprehensively review the mechanisms of DND1 protein and its biological roles. Especially, the potential involvement of DND1 in the etiology of somatic cancers. The manuscript is well organized and reasonably presented.

Minor problem. It will be much better If the authors draw a cartoon figure to describe the gene or protein structure in Page 2.

Reviewer 2 Report

In this manuscript the authors provide a nice review on the biological function of DND1, explore studies performed in somatic cancers and cell lines, and also describe TCGA data.

The article is well written and well structured.

I have the following comments:

  • The mechanism by which DND1 blocks access of microRNAs to mRNAs thus changing gene expression (especially of genes involved in regulation of apoptosis and cell cycle) is interesting. In the last decade the microRNAs of the miR-371-373 cluster have assumed the role of the most relevant biomarkers of TGCTs, and these have been shown to interact with p53 pathway by interfering with LATS2. Is there any link described between DND1 and this microRNA cluster?

  • The regulation of pluripotency and chromatin regulator genes mentioned in page 4, lines 128-132 is very interesting; authors could expand more this section, mentioning which genes are regulated. For instance, CRIPTO is a relevant player in regulation of germ cells and inappropriate activation may lead to TGCTs or on the other hand to infertility – any evidence that DND1 interferes with this pathway?

  • Authors mention that mutations in DND1 do not contribute sufficiently for TGCT incidence in humans. However, is there any study investigating protein expression of DND1 in human TGCT tissues and in human-derived GCT cell lines?

  • The same way, are there studies of DND1 expression in testicular parenchyma tissues of patients with different degrees of spermatogenesis?

  • In most studies in the somatic cancers downregulation of DND1 was found in the cancer tissues/cells. The exception was the colorectal cancer study – why?
  • Authors should expand more a Future perspectives section, where they elaborate on the advantages of pursuing research on DND1 and how it will contribute.

Round 2

Reviewer 2 Report

No further suggestions to the authors.